# Toward a Better Understanding of the Gelation Mechanism of Methylcellulose via Systematic DSC Studies

**DOI:** 10.3390/polym14091810

**Published:** 2022-04-28

**Authors:** Beata Niemczyk-Soczynska, Pawel Sajkiewicz, Arkadiusz Gradys

**Affiliations:** Institute of Fundamental Technological Research, Polish Academy of Sciences, Pawinskiego 5b St., 02-106 Warsaw, Poland; bniem@ippt.pan.pl (B.N.-S.); psajk@ippt.pan.pl (P.S.)

**Keywords:** methylcellulose, thermosensitive hydrogel, crosslinking, DSC

## Abstract

A methylcellulose (MC) is one of the materials representatives performing unique thermal-responsive properties. While reaching a critical temperature upon heating MC undergoes a physical sol-gel transition and consequently becomes a gel. The MC has been studied for many years and researchers agree that the MC gelation is related to the lower critical solution temperature (LCST). Nevertheless, a precise description of the MC gelation mechanism remains under discussion. In this study, we explained the MC gelation mechanism through examination of a wide range of MC concentrations via differential scanning calorimetry (DSC). The results evidenced that MC gelation is a multistep thermoreversible process, manifested by three and two endotherms depending on MC concentration. The occurrence of the three endotherms for low MC concentrations during heating has not been reported in the literature before. We justify this phenomenon by manifestation of three various transitions. The first one manifests water–water interactions, i.e., spanning water network breakdown into small water clusters. It is clearly evidenced by additional normalization to the water content. The second effect corresponds to polymer–water interactions, i.e., breakdown of water cages surrounded methoxy groups of MC. The last one is related to the polymer–polymer interactions, i.e., fibril hydrophobic domain formation. Not only did these results clarify the MC crosslinking mechanism, but also in the future will help to assess MC relevance for various potential application fields.

## 1. Introduction

Smart or so-called stimuli-responsive materials are up-and-coming to a wide range of scientific and industrial fields such as textiles, the food industry, sensors, or materials for biomedical applications [1,2,3,4,5,6]. Excellent examples of such materials are piezoelectric electrospun nanofibers [7], self-repairing films [8], or hydrogels [9]. The latter represents a unique behavior by changes in swelling, degradation, or gelation as a response to external stimuli in a specified and adjustable manner. In this respect, these materials might respond to pH, electric and magnetic field, light, or temperature [10,11,12]. Among many stimuli-responsive hydrogel materials, methylcellulose (MC) displays a thermal responsive character and deserves special attention due to its interesting physical crosslinking nature [13]. MC belongs to the simplest cellulose derivative, where hydroxyl groups (-OH), initially present in cellulose, are partially substituted with methoxy groups (-OCH_3_). Such modification makes cellulose amphiphilic, water-soluble, and exhibit thermally reversible crosslinking behavior near the physiological temperature, which is particularly interesting from the point of view of biopolymeric materials development [14,15].

MC aqueous solutions demonstrate physical crosslinking due to weak physical interactions which appear under particular temperature conditions. The lower critical solution temperature (LCST) defines the MC sol-gel transition. Depending on such parameters as substitution degree (DS), i.e., the average number of -OH groups substituted with -OCH_3_ in glucose units; the heating rate; the presence of additives; and molecular mass (MW), LCST might appear in the range of 30–80 °C [16]. Below LCST, MC aqueous solution is in a sol state, where solvent–solvent and polymer–solvent interactions dominate in the solution. Above LCST, polymer–polymer interactions start to prevail resulting in the formation of 3D hydrogel structure. In many literature reports, e.g., [17,18], describing LCST crosslinking behavior, it is stated that MC crosslinking occurs through several stages in low- as well as high-concentrated MC solutions. Nevertheless, the nature of MC crosslinking mechanisms is still under debate. According to Chevillard et al. [17], the MC multistep crosslinking mechanism, derived from rheological measurements, is explained by the existence of three gels in the phase diagram. The first one is a low-temperature low concentration gel which forms through weak hydrophobic domains, the second one was found at high concentration and forms through appearance of crystallites, while the third one forms during phase separation [19]. More current studies [18,20] tend to accept new MC crosslinking mechanisms based on fibril formation. This theory assumes primary nucleation followed by coalescence of associated chains with further crystallization. For many years, the MC crosslinking mechanism was related to destruction of “water-cages” surrounding -OCH_3_ groups and simultaneously interacting with -OH through hydrogen bonds followed by the formation of associated hydrophobic domains [21,22,23,24]. Other current studies of MC crosslinking mechanisms conducted by Yang et al. [25] confirmed this theory, but also showed that formation of associated hydrophobic domains of MC chains resembles fibril structures. The studies were confirmed with rheological measurements, accompanied with all-atom molecular dynamic simulations. Especially the latter method, as opposed to most of the experimental methods, allowed to detect and qualify the actual molecular interactions between MC chains and water molecules. 

The MC crosslinking mechanism still remains unclear because providing adequate experimental conditions is usually challenging. MC shows reversible sol-gel transition, and the thermally crosslinked hydrogel is expected to return to the sol form upon cooling to the temperature below LCST. Nevertheless, upon the temperature change, the sol-gel reversible transition kinetics might vary and be unrepeatable [11]. Those differences depend on heating/cooling rate, thus, on the time allowed for assembly of the hydrogel network or its decomposition. The sol-gel transition of MC has been characterized using various methods, e.g., rheological measurements such as dynamic mechanical analysis [26], the inversion tube [27], or DSC [19,28].

Rheological measurements might give clear information about materials’ crosslinking point when intersection of the storage modulus (G′) and the loss modulus (G″) curves results in clear dominance of G′ over G″ [29]. But there exist types of polymers, where crosslinking does not show any or clear enough the G′ and G″ intersection point. Some commercially available materials belong to this group including MC aqueous solutions. In such a case, depending on the measurement conditions: isothermal or in heating/cooling mode, detailed information on the maximum of crosslinking rate or crosslinking temperature might be determined from the time or temperature derivative of the G′ curve. Another difficulty in MC crosslinking measurements is ensuring hermetic conditions to avoid water evaporation resulting in unreliable results. Since rheological studies dedicated to hydrogels are based on plate or cone geometry, there are few methods to avoid water evaporation from the solution. One of them is using a solvent trap, e.g., silicone oil [30]. Nevertheless, in our previous studies [26], we observed that silicone oil partially reacted with MC hindering its crosslinking effect. Another method of avoiding water evaporation uses additional covering plates. Nevertheless, long measurements of MC crosslinking at higher temperatures results in partial drying out of MC solutions. In this regard, extra covering plates seem insufficient and although keeping the same parameters during measurements, obtained results might be biased.

Using the inversion tube method which ensures hermetic conditions, it is possible to indicate the sol-gel transition macroscopically determining mobility of crosslinked hydrogel after inverting the vial filled with the sample. When hydrogel does not flow after a particular time, e.g., 10 s, it is considered to be crosslinked [31]. However, that method is not able to provide fundamental information about hydrogels structure after crosslinking time, i.e., if the hydrogel crosslinked partially or formed a fully crosslinked polymeric network. The method also cannot provide any details of the crosslinking mechanism.

DSC seems to overcome all of the above-mentioned limitations connected with alternative sol-gel transition characterization methods. The hermetic environment preventing water evaporation might be easily ensured by using special hermetic pans, while registered thermograms may show the thermal effects accompanying the structural changes that take place in MC solution in an isothermal or heating/cooling mode. With this method, it could be clearly answered if MC crosslinking mechanism resembles more crystallites formation or hydrophobic associations, the first of which is exo- and the second- endothermic effect.

Therefore, our studies are aimed at clarification of the MC thermal crosslinking mechanism by thorough and systematical analysis of DSC thermograms for a wide range of MC concentrations. Our results show several endotherms appearing during heating as well as cooling, which have never been noticed and discussed before. The dependencies of thermal effects on MC concentration were used for the interpretation of molecular mechanisms during heating and compared with the experiments and interpretations described in the literature. These results allowed us to assess MC relevance for such applications as in situ crosslinking scaffolds for tissue regeneration, cell, growth factor, or drug delivery systems.

## 2. Materials and Methods

### 2.1. Materials

MC (METHOCEL A15LV, Sigma Aldrich, St. Louis, MO, USA, viscosity of 10–25 mPa*s, 2% in H_2_O (20 °C)) was used for hydrogel formation. A wide range of concentrations, i.e., 0.5–14 wt% was prepared according to the procedure reported in our previous publication [26].

### 2.2. DSC

Thermograms were registered using a power-compensation differential scanning calorimeter Pyris 1 DSC (Perkin-Elmer, Waltham, MA, USA). The scans were registered at non-isothermal conditions, at a constant heating/cooling rate of 2 K/min, in the temperature range from −5 to 100 °C, equilibrated isothermally for 5 min. The samples were loaded into dedicated stainless steel hermetically sealed pans that ensured no sample mass exchange with the environment, which was assured by checking the stability of sample mass. Preliminary measurements revealed a very low heat flow signal of the transitions; thus, special approaches were implemented. The pans were top filled resulting in the sample’s mass in the range of 63–78 mg. Instead of the standard approach of the measurements against an empty reference pan, in order to improve the heat flow signal from the thermal effects, the samples were measured against a reference pan filled with demineralized water of comparable mass c.a. 70 mg. This provided comparable heat capacities of the sample and reference. Additionally, the heating/cooling cycle was repeated ten times and averaged to increase the heat flow-to-noise signal ratio, which further improved the quality of measurements. The same procedures were used previously in [26], however, in slightly narrower temperature range, i.e., −10–80 °C and less MC concentration points have been studied. The choice to increase the low temperature limit from −10 to −5 °C enabled to avoid unwanted occasionally occurring crystallization of water in the reference pan. The choice to increase the high temperature limit from 80 °C to 100 °C substantially improved analysis of the thermal effects. In current studies we also decided to increase the number of MC concentration points to reveal and better understand the effect of the MC concentration on the crosslinking mechanism.

DSC heating/cooling scans registered at a constant rate were subjected to the analysis of the MC content impact on the thermal effects, i.e., on each of several over-imposed peaks. It was essential to separate individual peaks reflecting their asymmetric shape. Much effort was put into the peaks’ deconvolution (NLSF with tolerance 1 × 10^−9^, standard Leveberg Marquardt iteration algorithm, the confidence level for parameters 95%), which was realized using the asymmetric double sigmoid (ADS) function and the nonlinear least squares fitting method. Individual peaks were characterized by the parameters determined from ADS fitting and analysis as the peak area, representing the transition latent heat, ΔH, peak’s maximum temperature position, T_p_, and peak’s full width at half maximum, FWHM, reflecting the transition rate (all of the ADS function parameters are included in the Appendix A). Before peak deconvolution, the scans were subtracted with a baseline approximated with 5th order polynomial. All the data were analyzed using Origin software.

## 3. Results

Figure 1 shows the heating and cooling scans for all used MC concentrations after baseline subtraction and normalization to MC mass. Several thermal effects might be observed: all endothermic during heating and all exothermic during cooling. In the heating mode, there are, generally, two maxima; however, for the lower MC content (below 9 wt%) there is discernible a low temperature (LT) shoulder, which evidences its maximum only at MC 1 wt% (Figure 2). Although this shoulder was registered by Nishinary et al. [32], it was not analyzed. The MC 1 wt% curve after peak deconvolution showing evident three maxima is presented in Figure 2a and an example of a heating scan with the LT shoulder is provided for the MC 3 wt% in Figure 2b. In Figure 1a it may be observed that increasing the MC content leads to shifting of the peaks toward lower temperatures and to a decrease of the LT shoulder. It was found that above 9 wt% of MC, high quality of fitting was approached using only two peaks corresponding to the medium-temperature (MT) and the high-temperature (HT) transitions without taking into account the LT peak (Figure 2c). In our previous studies [26], for the low MC concentration range, instead of the LT, we reported a low temperature exothermic effect followed by two endothermic effects, i.e., MT and HT. This conclusion resulted from much lower the high temperature limit (80 °C), which affected the baseline subtraction. Thus, current results provide also an update of these previously published in [26].

Moreover, in the MC content range (2–2.5 wt%), a peculiarity was found, which is seen as a deviation in the thermal effects from the general trend. In order to confirm this behavior, several samples were investigated using new solutions prepared with additional MC concentrations between 2 and 2.5 wt%. In Figure 1a,b, there are these additional three curves with lower intensities, indicated as 2.05, 2.15, and 2.25 wt%, confirming a shift in peaks’ position, when compared to the general trend. The details of this peculiarity will be discussed further.

In the case of the cooling mode, in the whole MC content range, the exothermic effects always showed two maxima, and the use of two peaks in the peak deconvolution approach was found sufficient. An example of the peak deconvolution of the MC 1 wt% cooling curve is presented in Figure 2d. Regarding the difference in the number of the peaks observed during heating and cooling, the question, if one of the peaks is hidden or excluded from the cooling measurement, may be explained by comparing the total heat measured during both modes. In Figure 3, the total heats are presented as normalized to the sample mass (Figure 3a) and to the MC content mass (Figure 3b). It may be seen that both heats are very similar in value and trend; however, the heat upon cooling is higher by 14 wt% than the heat upon heating. It is generally encountered that the latent heat of transition (ΔH, enthalpy change) increases with temperature. Thus, when upon heating, the transition takes place at a higher temperature than upon cooling, the heat upon heating would be expected higher. The reason behind the higher cooling heat value is not clear at the moment; however, from similar values and trends it may be rather expected that the heat of the LT peak, as observed during heating, is present but hidden in the thermal effects observed during cooling.

Moreover, in Figure 3a it is seen that the ΔH normalized to the sample mass, depends on MC content quite linearly. For cooling mode, there is a higher slope (0.16 J/g_MC_) than for heating mode (0.12 J/g_MC_); however, both dependencies do not extrapolate to zero at zero MC content but lead to similar ordinates, c.a. 0.213 J/g. It suggests that some of the heat may come from another source than MC, most probably from water. Another indication of this observation comes from Figure 3b presenting the transition heat normalized to the MC mass. It may be seen that in the lower MC content range, the values are very high reaching 45 and 50 J/g_MC_ and the heats decrease strongly before reaching plateaus at c.a. 15 and 18 J/g_MC_ for heating and cooling, respectively, which start from c.a. 7 wt% of MC. This constant transition heat indicates its dependence on MC only.

Further detailed analysis of the thermal effects relies on results obtained using the peak deconvolution approach. The results are presented as a function of MC content providing the peak parameters such as the temperature peak position (T_p_), the ΔH, and the peak full width at half maximum (FWHM) in Figure 4 for heating and Figure 5 for cooling. With the increase of MC content, all the peaks’ positions, generally, shift to a lower temperature (Figure 4a and Figure 5a); however, the LT peak position in heating mode (Figure 4a) is the least affected. In Figure 4b, presenting the peaks’ heat normalized to the sample mass, the heat of the MT and HT peaks generally increases dynamically with the MC content, while the LT peak heat shows a slow decrease. These observations indicate that the MT and HT peaks relate to the transitions involving MC molecules, while the LT peak relates to a transition involving water molecules only. This conclusion is supported by the domination of the LT peak over the MT and HT peaks in the lower MC concentration range. The LT peak heat being c.a. 0.22 J/g is similar to the value obtained with a linear approximation of the total heat in Figure 3a, decreases slowly to c.a. 0.15 J/g in the MC content range 6–8 wt% and above 8 wt% the peak is not detectable.

Another parameter analyzed is the peak’s width, FWHM, describing the temperature range of the transition, which might be related to the transition rate. Thus, a higher transition rate could be expected for a narrow peak. It is shown in Figure 4c that the strongest changes with MC content are seen for the MT peak showing a four-fold increase. The widths of the LT and HT peaks were found the highest and lowest, respectively, and weakly dependent on MC content, except for a decrease at the lowest MC content in the case of the LT peak and a small increase at the highest MC content in the case of the HT peak.

Clear evidence that the LT peak’s heat is probably related to water molecules only, and the heats of the MT and HT peaks are related to MC molecules, comes from the comparison of the heats using two different normalizations—to water and MC content (Figure 4d). It is seen that the LT peak’s heat increases with the increase of water content (decrease of MC content) approaching the extrapolated value at zero MC content, which is practically the same irrespective of the type of normalization (Figure 3a and Figure 4b,d).

In the case of MT and HT heats in J/g_MC_ (Figure 4d), there is a relatively large deviation in the lower MC content range, up to 7 wt%, making the analysis more difficult. In this MC content range, the heat dependencies for both MT and HT peaks can be treated as more or less independent on MC content with c.a. 10 J/g_MC_ and 2.6 J/g_MC_ for the MT and HT peak, respectively. In the higher MC content range, above 7 wt%, both peaks show opposite behavior characterized by local extrema at 11 wt%. The heat of the MT peak reaches 13.4 J/g_MC_ at the maximum, and the heat of the HT peak decreases to 1.9 J/g_MC_ at the minimum.

In the cooling mode (Figure 5a), the temperature positions of the two peaks, MT and HT, follow similarly decreasing trends as in the heating mode (Figure 4a). The heats of both peaks normalized to sample mass increase with MC concentration, except for the last point (Figure 5b). Moreover, both heats extrapolated to zero MC content result in the same heat, c.a. 0.1 J/g (Figure 5b). In the case of the heats normalized to MC content (Figure 5d), in the lowest MC content range, both dependencies decrease steeply with MC content. These behaviors indicate that the LT peak as observed upon heating is included in the heat registered upon cooling. Furthermore, upon cooling there is an opposite and much different relation of the two heats than observed upon heating for the heats of the MT and HT peaks. First, upon cooling the HT peak, heat dominates over LT by 5 wt% (Figure 5b), while upon heating the MT peak, heat is four to seven times higher than the HT peak heat (Figure 4b). This indicates that comparing the two modes, the transitions rather proceed using different routes and mechanisms. Thus, the LT peak hidden under the cooling peaks treated as the MT and HT peaks makes the analysis not clear.

Regarding the peak width, FWHM, in cooling mode (Figure 5c), its dependence on MC content is more complex than that observed for heating mode (Figure 4c). The MT peak dependence increases strongly with the MC content, while the HT peak dependence is much flatter.

The peculiarity in the MC content range (between 2 and 2.5 wt%), was found related to changes in the MT peak. It clearly manifests as an increase in temperature position, T_p_, an increase in the peak’s width, FWHM, and a decrease in the transition heat, ΔH, observed in both the heating and cooling modes (indicated by vertical sticks in Figure 4 and Figure 5). The peculiarity is most probably related to the slowing down of the MT transition rate. A deeper explanation of this phenomenon needs further investigation.

## 4. Discussion

A molecular interpretation of the phenomena during gelling of MC solutions based on the DSC results is given below. First, the endotherms visible on DSC scans might be an effect of polydispersity of MC molecular weight, heterogeneity of the MC substitution degree (SD) or inhomogeneous position of the -OCH_3_, and multistep mechanism of MC crosslinking [32]. According to many literature reports, e.g., [18,20,25,33,34,35,36,37,38] we will discuss the last reason for the appearance of multiple effects of MC gelation.

The lack of exothermic effects during heating does not support the mechanism of the primary nucleation and subsequent crystals growth as proposed by Coughlin or Schmidt [18,20]. Since crystals formation is accompanied by exothermic effects, that are not revealed in our current studies, we dismiss this theory.

On the contrary, in line with our DSC results is the interpretation of the MC crosslinking mechanism based on the water cages’ destruction and association of the fibril hydrophobic domains. Our interpretation is that the first transition manifested by the LT endotherm at c.a. 50 °C, is an effect of water–water interactions close to polymer chains, while the second-MT endotherm at c.a. 55–70 °C, and the third-HT one at c.a. 65–72 °C, correspond to the polymer–water and polymer–polymer interactions, respectively. The LT endotherm can be explained by the so-called thermal breaking of the hydration shell described widely in the literature [33,34,35,36]. Briefly, two different states of hydration water can coexist simultaneously at lower temperatures, i.e., shells of water formed around amphiphilic polymers. It is recalled that the amphiphilic character of MC results from the presence of both the hydrophilic -OH and the hydrophobic -OCH_3_ groups in its molecular chain. One state of water is characteristic for low-concentrated solutions, where large water aggregates surrounding polymer molecules appear using hydrogen bonds. According to Brovchenko et al. [33], in this state, called the spanning water network, hydrated structures are more ordered than that of bulk water. It was explained by molecular dynamics simulations [37] that the water dipole moment becomes oriented with polymer structure due to its much slower relaxation process. The second state of hydration water is characteristic for higher concentrated solutions, where small water clusters surround polar and nonpolar polymer groups. The transition from one state to the other takes place during heating and has been described as the thermal breaking of the hydration shell [36]. In detail, the process takes place, when the dominating spanning water network breaks down to form more disordered small clusters, which is a result of decreasing number of hydrogen bonds broken by temperature increase. This phenomenon is observed at c.a. 50 °C, which corresponds to the first endotherm observed in Figure 2a. From the literature, we know the H-bond rupture processes only, in which energy according to molecular simulation ranges from 0.2 to 4.2 kcal/mol [36]. According to our results, thermal breaking of the hydration shell might cease from c.a. 9 wt% (Figure 2c), meaning only small water clusters exist in the solution. The MT endotherm is interpreted as coming from polymer–water interactions and is an effect of dehydration of water from water cages that surround -OCH_3_ groups and destruction of hydrogen bonds between water molecules and -OH groups in MC [38]. The confirmation of the described above theory found an additional confirmation in studies conducted by Yang et al. [25], where it was shown that the number of hydrogen bonds between MC chains and water molecules significantly decreases with increasing temperature.

After these processes, which correspond to the first stage of MC crosslinking, the MC chains start to reorganize forming intra- and intermolecular MC–MC hydrogen bonds and MC–MC hydrophobic interactions. Yang et al.’s analysis [25] of hydrophobic and hydrophilic solvent-accessible surface area (SASA) showed that at higher temperatures the contribution of hydrophobic interactions prevails over hydrogen bonding in the solution, resulting in MC chains aggregation. Bodvik et al. [39] explain that MC chains are arranged in fibril structures, to minimize the energy of the hydrogel system by a maximum decrease of the contact between -OCH_3_ groups with water molecules. This process is observed as the HT endothermic peak.

Reproducibility of the thermal effects during several repetitive heating and cooling cycles proves reversible character of MC crosslinking process (Figure 2d). The HT exotherm observed during cooling may correspond to massive dissociation of fibril hydrophobic aggregates with simultaneous rearranging of water molecules into more ordered structures. As a result, the fibril network is gradually weakened [29]. Since the two processes occur together, more heat is exchanged resulting in the dominance of HT exotherm over MT [40] (Figure 5b,d). According to Li et al. [29], the LT exotherm occurs at the critical temperature, at which the hydrogel network has been completely interrupted. The LT exotherm is related to the formation of water molecules around nonpolar regions of MC polymeric chains also known as water cages and continues the formation of water–MC and water–water hydrogen bonds. These processes might occur simultaneously, since they are visible as one exothermic peak.

According to the current report of Bonetti et al. [11], but also the previous ones [41,42,43], an increase in MC concentration in the solution leads to a decrease in LCST. The higher MC concentration leads to an increase in the density of the polymer network in the solution leading to enhanced polymer–polymer interactions at lower temperatures [11,43,44]. It is the result of the decreased contribution of interaction between water–water over water–polymer and polymer–polymer interactions with the increased MC concentration (Figure 4a). The fact that for low MC concentrations, the LT and MT consume more heat than HT during heating (Figure 4b) may be explained by a large amount of energy needed to destroy strong hydrogen bonds between water molecules. This process is observed as LT. A lot of energy is also used to break the water cages surrounding -OCH_3_ groups, resulting in a prevailing endothermic effect (MT). Li et al. [29] reported during heating most of the heat is used to destroy hydrogen bonds between water molecules and water cages. The remaining heat is used for hydrophobic aggregation which is registered as HT. Li et al. showed that the heat needed for the formation of hydrophobic aggregates (observed as HT effect) is always lower than that needed for water cages breakdown (observed as LT effect), which is also observed in Figure 4b.

While heats are normalized to H_2_O and MC mass, similarly to the normalization to the sample’s mass, the LT effect decreases to 0 with MC contribution. The polymer phase contribution prevails over the solvent and there are diminished amounts of hydrogen bonds between water molecules. 

The decrease of MT and HT exotherm heats during cooling with increasing MC concentration (Figure 5d) might be explained as follows. During cooling, the hydrophobic fibril network is decomposed and simultaneously water molecules start to form ordered structures. These two processes are visible as one HT exotherm. More heat is released during water molecules organization (formation of strong hydrogen bonds between water molecules) than by dissociation of weak hydrophobic interactions. While the increase of MC concentration leads to formation of fewer hydrogen bonds between water molecules, resulting in a significant decrease of released heat. The MT exotherm corresponds to water cages formation and further hydrogen bonds formation. The decrease of MT transition heat with MC increasing concentration has a similar reason as in the case of HT where fewer hydrogen bonds are formed at lower amounts of water molecules.

## 5. Conclusions

In our research, we justified the gelation mechanism of MC through systematic investigations of a wide range of MC concentrations using DSC measurements. The results prove the MC gelation is a multistep reversible process dictated by the LCST character. The gelation occurring during heating is manifested by three or two endotherms, depending on more or less diluted MC solutions, respectively. An additional first endothermic effect observed for lower concentrated MC solutions has not been described so far. It was evidenced by our results that this low-temperature effect corresponds to the interactions between water molecules, i.e., destruction of the spanning water network formed by hydrogen bonding into small water clusters. The other further two effects are related to polymer–water, which is destruction of “water cages” around -OCH_3_ groups, and polymer–polymer interactions that is the formation of fibril-like hydrophobic domains.

We believe that our results allow a comprehensive understanding of the MC gelation mechanism and will be useful for further studies related to MC characteristics and designing MC-based hydrogel systems for a wide range of potential applications such as tissue engineering, drug-, cell-, growth factors delivery, and diagnostics.

## Figures and Tables

**Figure 1 polymers-14-01810-f001:**
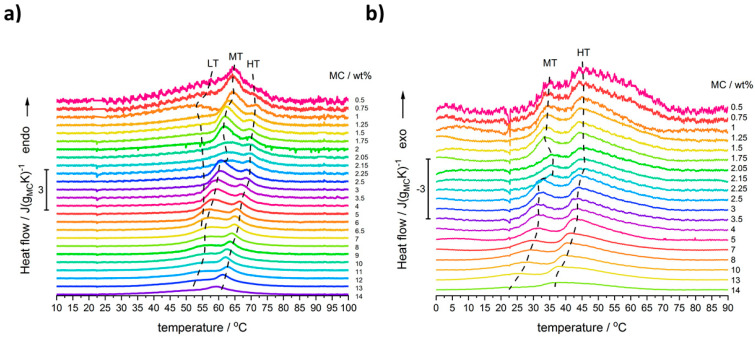
DSC scans registered during: (**a**) heating and (**b**) cooling for solutions of MC at mass content as indicated, normalized to MC mass. Curves shifted in Y-axis for clearness.

**Figure 2 polymers-14-01810-f002:**
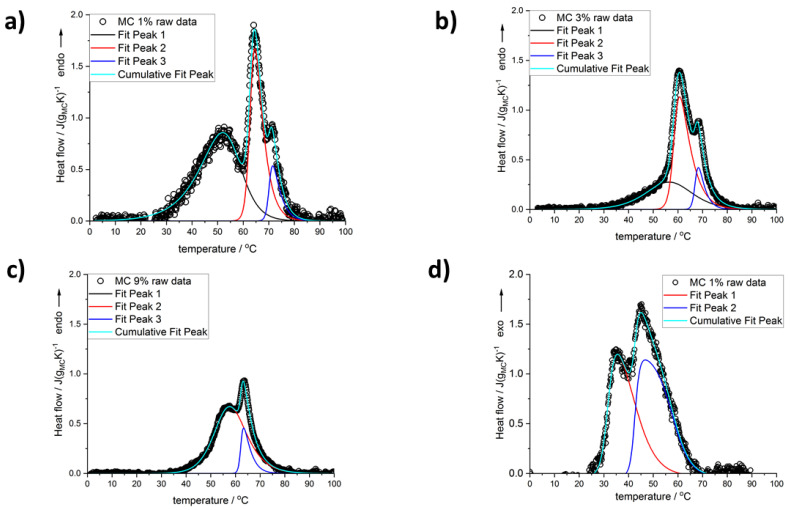
Deconvolution of the peaks seen on the scans registered for MC content: (**a**) 1 wt%, (**b**) 3 wt%, (**c**) 9 wt% during heating, and (**d**) 1 wt% during cooling.

**Figure 3 polymers-14-01810-f003:**
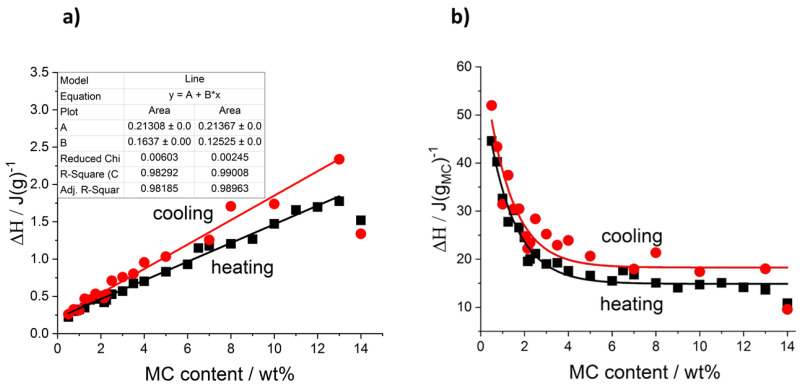
Total transition heat determined during heating and cooling: (**a**) normalized to the sample mass, (**b**) normalized to the MC content mass.

**Figure 4 polymers-14-01810-f004:**
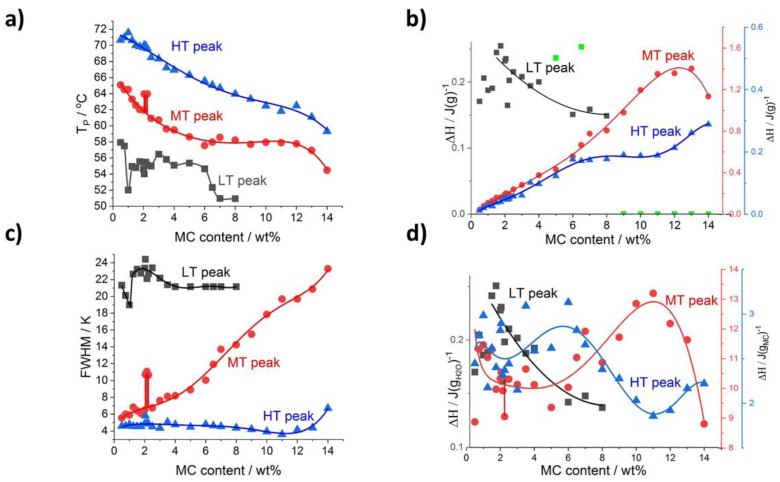
Deconvolution results for heating scans as a function of MC content: (**a**) peaks’ temperature position and (**b**) peak’s transition heat normalized to sample mass (**c**) peaks’ width FWHM, (**d**) peak’s heat normalized to H_2_O (LT peak) and MC content (MT and HT peaks). Green points in (**b**) are LT peak area values excluded from the trend analysis due to high deviation.

**Figure 5 polymers-14-01810-f005:**
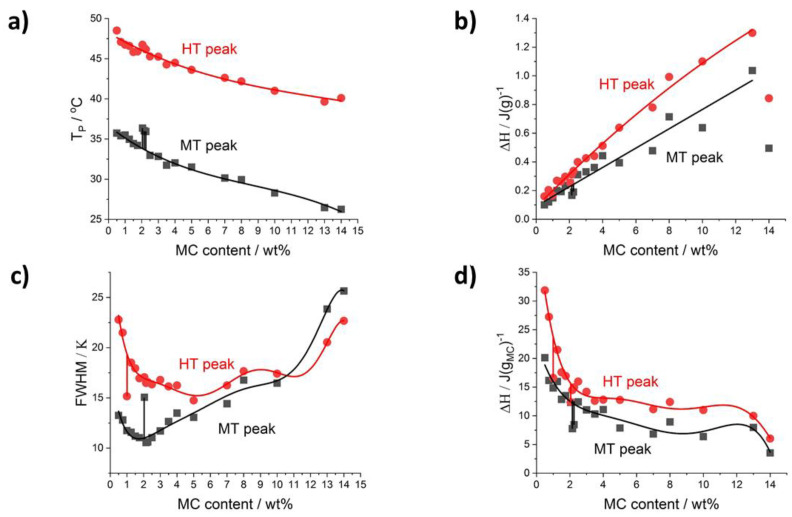
Deconvolution results for cooling scans as a function of MC content: (**a**) peaks’ temperature position, (**b**) peak’s transition heat normalized to sample mass, (**c**) peaks’ full with at half maximum, (**d**) peak’s heat normalized to MC mass.

## Data Availability

The data used in this article can be retrieved by private corresponding author request.

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
