# Peer review of "Toward a Better Understanding of the Gelation Mechanism of Methylcellulose via Systematic DSC Studies"

_polymers, 2022, doi:10.3390/polym14091810_

Round 1
Reviewer 1 Report
The manuscript entitled "Towards a better understanding of the gelation mechanism of methylcellulose via systematic DSC studies" has been reviewed. The results are very helpful. However, the manuscript needs to be well revised before acceptance. Detailed comments are as follows:
- There are some typo errors in the manuscript. Some numbers and letters, e.g., 3 in -OCH3, MC in gMC and p in Tp, should in the subscript form. In Line 174, 3wt. % should be 3 wt. %. Please spell-check the whole manuscript.
- The abbreviations, such as DSC, LT, FWHM and delta H, should not be defined more than once.
- In Abstract, the full name of MC is methylcellulose, not methylcellulose aqueous solution. “these studies” should be “this study”.
- Correct unnecessary capitalization of first letters of some phrases in both Abstract and main text, such as Lower Critical Solution Temperature, Differential Scanning Calorimetry, Asymmetric Double Sigmoid, and Nonlinear Least Squares Fitting.
- In Line 46, molecular mass (Mw) should be molecular weight (MW). In polymer science, Mw represents to weight-average molecular weight.
- In 2.1. Materials, detailed information on MC, such as molecular weight, should be provided.
- In all DSC thermograms, heat flow should be Heat flow.
- In Fig. 1, the unit of y-axis should be a.u. since every DSC thermogram had been vertically translated.
- In Line 180, the unit of J/(gMC %)) is not correct.
- In Figs. 4b and d, the y-axis names are too close to the axis.
- Please unify wt.% and wt%. In addition, wt. % should be wt%.
- In references, the last one is a journal, not a book.
Author Response
The manuscript entitled "Towards a better understanding of the gelation mechanism of methylcellulose via systematic DSC studies" has been reviewed. The results are very helpful. However, the manuscript needs to be well revised before acceptance. Detailed comments are as follows.
Answer: Dear Reviewer,
Thank you for your valuable comments and suggestions. We did our best to improve the manuscript’s quality. According to your comments, we have made corrections in the manuscript. Our answers and changes done in the text are listed below.
- There are some typo errors in the manuscript. Some numbers and letters, e.g., 3 in -OCH3, MC in gMC and p in Tp, should in the subscript form. In Line 174, 3wt. % should be 3 wt. %. Please spell-check the whole manuscript.
Answer: The corrections have been done.
- The abbreviations, such as DSC, LT, FWHM and delta H, should not be defined more than once.
Answer: The corrections have been done.
- In Abstract, the full name of MC is methylcellulose, not methylcellulose aqueous solution. “these studies” should be “this study”.
Answer: The corrections have been done.
- Correct unnecessary capitalization of first letters of some phrases in both Abstract and main text, such as Lower Critical Solution Temperature, Differential Scanning Calorimetry, Asymmetric Double Sigmoid, and Nonlinear Least Squares Fitting.
Answer: The corrections have been done.
- In Line 46, molecular mass (Mw) should be molecular weight (MW). In polymer science, Mw represents to weight-average molecular weight.
Answer: The correction has been done.
- In 2.1. Materials, detailed information on MC, such as molecular weight, should be provided.
Answer: The manufacturer has not provided the molecular weight of METHOCEL A15LV. Instead of molecular weight, we provided in the manuscript information on the viscosity of 10-25 mPa.s, 2 % in H2O (20 °C).
- In all DSC thermograms, heat flow should be Heat flow.
Answer: The correction has been done.
- In Fig. 1, the unit of the y-axis should be a.u. since every DSC thermogram had been vertically translated.
Answer: The correction has been done. Since the proposed by the Reviewer arbitrary units could be misleading, we decided to keep them. However, instead of the Y-axis, we provided a scaling bar to make Fig. 1 more correct.
- In Line 180, the unit of J/(gMC %)) is not correct.
Answer: The correction has been done.
- In Figs. 4b and d, the y-axis names are too close to the axis.
Answer: The corrections have been done.
- Please unify wt.% and wt%. In addition, wt. % should be wt%.
Answer: The corrections have been done.
- In references, the last one is a journal, not a book.
Answer: The correction has been done.
Reviewer 2 Report
Recommendation: Minor revisions are needed.
Comments:
The paper by Niemczyk-Soczynska et al. contributes to the aspect of the MC gelation mechanism through examination of a wide range of MC concentrations via DSC. The title and abstract are appropriate for the content of the text. Moderate English changes are required. The article gives an interesting historical and scientific perspective on methylcellulose.
Some issues should be addressed before publication.
- Page 1, line 38, line 45. -OCH3
- Figure 1. For more accessible elucidation, please label the critical temperature and significant peak change on your DSC graph; since you have a multi-overlap chart, you can mark the peak on the top of the first DSC curve, such as MC/ wt% =0.5. For example, temperature= 22 ℃ for both graphs a and b.
- Figure 2. What is the peak deconvolution model used in this analysis? If it was not discussed, please add this information to the experimental method section.
- Figure 3. Please adjust the font size in image a (equation details).
Author Response
The paper by Niemczyk-Soczynska et al. contributes to the aspect of the MC gelation mechanism through examination of a wide range of MC concentrations via DSC. The title and abstract are appropriate for the content of the text. Moderate English changes are required. The article gives an interesting historical and scientific perspective on methylcellulose.
Some issues should be addressed before publication.
Answer: Dear Reviewer,
Thank you for your valuable comments and suggestions. We did our best to improve the manuscript’s quality. According to your comments, we have made corrections in the manuscript. Our answers and changes done in the text are listed below.
- Page 1, line 38, line 45. -OCH3
Answer: The corrections have been done.
- Figure 1. For more accessible elucidation, please label the critical temperature and significant peak change on your DSC graph; since you have a multi-overlap chart, you can mark the peak on the top of the first DSC curve, such as MC/ wt% =0.5. For example, temperature= 22 ℃ for both graphs a and b.
Answer: The corrections have been done. We provided guiding lines indicating the positions of the critical temperatures labeled as LT, MT, and HT in Figures 1 a and b.
- Figure 2. What is the peak deconvolution model used in this analysis? If it was not discussed, please add this information to the experimental method section.
Answer: NLSF with tolerance 1e-9, standard Leveberg Marquardt iteration algorithm, the confidence level for parameters 95 %. We added this information to the text.
- Figure 3. Please adjust the font size in image a (equation details).
Answer: The corrections have been done.
Round 2
Reviewer 1 Report
The manuscript has been well revised. It can be accepted if the following comments are considered:
- The number in H2O should in the subscript form.
- mPa.s should be mPa·s.
- Space should be put between number and wt%.
Author Response
All of the corrections have been done.